# Spike Timing-Dependent Plasticity at Layer 2/3 Horizontal Connections Between Neighboring Columns During Synapse Formation Before the Critical Period in the Developing Barrel Cortex

**DOI:** 10.3390/cells14181459

**Published:** 2025-09-18

**Authors:** Chiaki Itami, Fumitaka Kimura

**Affiliations:** 1Department of Physiology, Faculty of Medicine, Saitama Medical University, Moroyama, Saitama 350-0495, Japan; 2Laboratory of Brain Neuroscience, Faculty of Health Care Sciences, Jikei University of Health Care Sciences, Osaka 532-0003, Japan

**Keywords:** STDP, mouse, L2/3-L2/3, neighboring columns

## Abstract

The Hebbian type of spike timing-dependent plasticity (STDP) with long-term potentiation and depression (LTP and LTD) plays a crucial role at layer 4 (L4) to L2/3 synapses in deprivation-induced map plasticity. In addition, plasticity at the L2/3 horizontal connection is suggested to play an additional role in map plasticity, especially for “spared whisker response potentiation.” Unimodal STDP with only LTP, or all-LTP STDP drives circuit formation at thalamocortical, as well as L4-L2/3 synapse before the critical period. Here, we first show that the L2/3 horizontal connections exhibit all-LTP STDP when axons are extending during synapse formation before the critical period. LTP-STDP induced by pre-post timing was mediated by NMDA-R because APV blocked the induction. In addition, PKA signaling was involved because PKI 6-22 blocked the induction. However, LTP-STDP induced by post-pre timing was not mediated by NMDA-R, because APV could not block its induction. Nevertheless, PKA signaling was also involved in its induction because PKI 6-22 blocked the induction. Our finding indicates that PKA signaling plays an important role in all-LTP STDP during synaptic formation at the L2/3-L2/3 connection between neighboring columns with a distinct source of Ca^2+^ influx in the developing mouse barrel cortex.

## 1. Introduction

Rodent primary somatosensory cortex (S1) provides an excellent model system for studying the mechanism of plasticity in which sensory maps are highly plastic. In S1 map plasticity, the differential use of two sensory inputs changes cortical representation: an expansion of representation occurs for overused inputs and shrinkage for underused inputs. Recently, substantial progress has been made in identifying the underlying cellular mechanisms for map plasticity in S1, which has been considered to have at least two separable components [1,2]. The first component is a selective weakening of neural responses to deprived whiskers. The cellular basis of this process is thought to be the long-term depression (LTD) component of Hebbian STDP at L4–L2/3 synapses [3,4], where intact sensory inputs from the receptive field whiskers cause L4 followed by L2/3 spiking, leading to long-term potentiation (LTP), while in the deprived column, reversed spiking of L2/3 followed by L4 neurons occurs, leading to LTD. The second component of map plasticity is an enhancement of responses to spared whiskers in the deprived columns [5,6,7], which has been suggested to result from enhanced excitatory synaptic connection among cross-columnar projections from spared to deprived columns. A strong cellular candidate for this process would be L2/3 horizontal connections [1,2,8,9]. In fact, the functional reorganization of horizontal L2/3 to L2/3 has been observed in the rodent L2/3 [10,11,12] and cat L2/3 [13] of adult animals but not in the somatosensory cortex of developing rodents. Recently, the exact ages of transition from neonatal network formation to the initiation of the critical period were hypothetically proposed [2]. Interestingly, during network formation before the critical period, a distinct type of STDP, or all-LTP STDP, operates, in which both pre-before-post and post-before-pre timing cause LTP in a spike timing-dependent manner, that is, the shorter the delay, the larger the potentiation. This type of STDP is suitable for strengthening synaptic connection during synapse formation, when the spiking order of pre- and postsynaptic cells is still unstable or even random due to weakness in immature synapses. Thus, the developmental switches of STDP were observed from network formation to the initiation of the critical period at thalamocortical and L4-L2/3 synapses [2,4,14].

In the present study, we tried to determine whether STDP could be induced at horizontal L2/3 connections across columns before the suggested age of the initiation of the critical period, that is, before P15. As mentioned above, previous studies have demonstrated that all-LTP STDP appears during network formation; it is highly intriguing to determine whether the same mechanism mediates the formation of L2/3–L2/3 horizontal connections. Pyramidal cells in L2/3 have extensive horizontal projections directing neighboring columns. The developmental study demonstrated that horizontal projections are limited at P7 but extend massively around P10 to P14/P15 and subsequently continue to extend through the observation period up to P21/P22 [15]. It has been suggested that robust map plasticity starts at around P13–15 after the STDP at L4–L2/3 synapses switches from all-LTP STDP to Hebbian STDP with LTP and LTD [2,4], following the maturation of parvalbumin GABA neurons which enable the sequential L4 followed by L2/3 spiking in response to sensory inputs [2,16,17]. Thus, we tested whether the L2/3 horizontal connection between neighboring columns exhibits all-LTP STDP from P9 to P13 during the period of network formation before the initiation of the critical period. We found that this connection exhibits all-LTP STDP, although with a distinct induction mechanism between the inducing stimulations of pre-before-post and post-before-pre timing, both sharing PKA signaling but with distinct involvement of NMDA receptors.

## 2. Materials and Methods

The experimental protocols used in this study were approved by The Animal Welfare Committee of the Saitama Medical University (Approval Code: #4241).

### 2.1. Slice Preparation for Electrophysiology

Mice (C57BL6/J) were deeply anesthetized with Isoflurane (Pfizer, New York, NY, USA) and the whole brain was rapidly removed and transferred to ice-cold artificial cerebrospinal fluid (ACSF), consisting of (mM) 124 NaCl, 3 KCl, 1.2 KH_2_PO_4_, 1.3 MgSO_4_, 2 CaCl_2_, 10 glucose, 26 NaHCO_3_, bubbled with 95% O_2_, and 5% CO_2_ and balanced at pH 7.4 (295–305 mOsm). As described previously [18], thalamocortical slices (350–400 µm) were prepared from postnatal day 9 (P9) to P13 using a previously described thalamocortical sectioning procedure with rotor slicer, which was modified from an original sectioning procedure with vibratome [19]. Slices were immediately transferred to a holding chamber, submerged in oxygenated ACSF for at least 1 h, and then transferred to a recording chamber (27–30 °C) on the stage of an upright microscope (Olympus, BX51WI, Tokyo, Japan).

### 2.2. Electrophysiology

Whole-cell patch pipettes (5–10 MΩ) were used to record the membrane voltage from visually identified pyramidal cells in L2/3, as described previously [17]. Micropipettes were pulled from borosilicate thick-walled glass capillary tubing (Sutter Instruments, Novato, CA, USA). The pipette solution contained, in mM, 130 K-methane sulfonate, 10 KCl, 10 HEPES, 0.5 K-ethylene glycol tetraacetic acid, 5 Mg-adenosine triphosphate, 1 Na-guanosine-5′-triphosphate, 10 Na-phosphocreatinine, and pH 7.3 (295 mOsm). Responses were recorded using an Axoclamp 2B amplifier (Molecular Devices, Palo Alto, CA, USA) in current-clamp mode, unless otherwise specified. Signals were low-pass-filtered at 3–5 kHz, digitally sampled at 10–20 kHz, and monitored with the pCLAMP 9 software (Molecular Devices). Upon inserting the electrode into the bath, the stray pipette capacitance was compensated, and so was the bridge balance through a built-in circuit of the amplifier. The bridge balance was checked repeatedly and readjusted if necessary. A giga-ohm seal was made in voltage-clamp mode; then, once whole-cell recording was established, the recording mode was moved to the current clamp. Only excitatory regular spiking cells were included in the current study, identified by the responses to graded current injections. In response to the electrical stimulation, excitatory postsynaptic potentials (EPSPs) were recorded, and monosynaptic nature was first confirmed by short (typically < 2 ms from L2/3 stimulation), constant latencies, following 1 Hz afferent stimulations without failure. Unless otherwise specified, a concentric bipolar stimulating electrode (Frederick Haer & Co, Bowdoin, ME, USA) was placed in L2/3 of the neighboring column through which electrical stimuli consisting of square pulses of 100 μs up to 0.5 mA were applied every 7500 ms (0.133 Hz), including control as well as pairing stimulations. L2/3 neurons send lateral branches out to neighboring columns, spanning two to seven columns in adults [20]. Since we worked on animals of early ages during development, starting to extend their axons, and we adopted responses with short and constant latencies (see above); thus, our recordings are likely to consist of mostly from cells in the next neighbor columns, although we could not rule out the possibility of containing those from passing fibers of cells in the further columns.

### 2.3. Induction Protocol for STDP

Single postsynaptic action potentials were evoked by depolarizing current injection to the soma with the smallest current possible (typically < 1.5 nA for 5–10 ms). The stimulus intensity to presynaptic cells in L2/3 was also adjusted to evoke EPSPs with a single component without notches in the rising or decaying phases. Cells with changes in the resting membrane potential less than 2.5 mV (<±5 mV) were adopted. EPSPs with changes in peak amplitudes less than 10% for at least 5 min were further applied with the spike timing-dependent pairing stimulations. The pairing interval was defined from the onset of EPSPs to the peaks of each action potential. A total of 90 pairings were applied to presynaptic and postsynaptic cells with fixed delays. Stimulation patterns were constructed using a custom program written by LabVIEW 2012 (National Instruments, Austin, TX, USA), run on a Windows computer, and delivered through an interface (USBX-I16P, Technowave, Tokyo, Japan) from the stimulator.

### 2.4. Drugs

NMDA receptor antagonist, D-(-)-2-amino-5-phosphonopentanoic acid (D-AP5) was obtained from Tocris Bioscience (R&D Systems/Techne, Tokyo, Japan). Protein kinase A (PKA) inhibitor, 6-22 amide (PKI6-22), was obtained from Calbiochem (Merck/EMD, Tokyo, Japan). D-AP5 was applied from the bath, and PKI6-22 was included in the recording patch pipettes.

### 2.5. Statistical Analysis

All values in the text and figures are presented as mean ± SEM unless otherwise specified. Student’s unpaired *t*-tests and one-way ANOVA analysis were performed using a statistical software JMP14. A *p*-value of <0.05 was considered statistically significant.

## 3. Results

### 3.1. L2/3 Horizontal Connections to Adjacent Column Exhibit All-LTP STDP After P9

In the present study, since we are interested in the STDP during synapse formation before the initiation of the critical period, we tested the induction of STDP from P9 to P13. EPSPs were recorded from L2/3 pyramidal neurons in response to the stimulation applied to the adjacent column (Figure 1A). Throughout the period, monosynaptic, short-latency (typically < 2 ms) EPSPs were recorded. We tested various timing differences in pre-before-post and post-before-pre stimulation from +200 ms (pre-before-post) to −200 ms (post-before-pre) timing of stimulation. We found that both pre-before-post as well as post-before-pre stimulation caused LTP in a timing-dependent manner, that is, the shorter the delay, the larger the magnitude, although irrespective of the order of pre- or postsynaptic activity, which we call all-LTP STDP [2]. Examples of LTP are shown in Figure 1B,C for pre-before-post and post-before-pre timing, respectively. The relationship between magnitude of LTP and applied stimulus timing is shown in Figure 1D. Significant (*p* < 0.001) LTP was seen from +25 to −25 ms of delays at P9–P13 (Figure 1D). These experiments indicate that L2/3 neurons projecting to the adjacent barrel columns start to exhibit clear all-LTP STDP at least from P9, during synapse formation before the suggested age of the initiation of the critical period of map plasticity in the barrel cortex.

### 3.2. LTP-STDP by Pre-Before-Post Timing Stimulation Requires NMDA Receptor and PKA

Earlier studies showed that LTP requires NMDA receptor (NMDA-R)-mediated rise in intracellular Ca^2+^ at the postsynaptic site, which in turn activates calcium/calmodulin-dependent protein kinase II (CaMKII) in a rather matured cortex [21,22,23,24,25,26].

In the developing barrel cortex [4], during synapse formation, Ca^2+^ influx through NMDA-R is required, but activation of PKA instead of CaMKII is required in all-LTP STDP at L4-L2/3 synapses [4]. Thus, we examined the involvement of NMDA-R and PKA activation in all-LTP STDP at L2/3 between adjacent column connections. We first focused on the pre-before-post timing stimulation. We found that NMDA-R was required for LTP by pre-before-post timing stimulation, because the NMDA-R antagonist, D-AP5, at the concentration of 50 μM, blocked the induction of LTP-STDP, as exemplified in Figure 2A. Similarly, LTP-STDP was blocked by PKI6-22 (20–40 μM), a PKA inhibitor, as shown in Figure 2B. The quantification of the population data for D-AP5 and PKI6-22 is shown in Figure 2C. In the presence of D-AP5 and PKI 6-22, EPSPs following pre-before-post timing stimulation became 100.0 ± 3.2% (n = 7) and 98.1 ± 2.8% (n = 13), respectively, compared to the control group (121.6 ± 6.1%, n = 22). One-way ANOVA analysis indicated significant differences among groups (F(2, 39) = 5.84, *p* = 0.0063). The following post hoc test indicated that the differences between control, D-APV, and PKI 6-22 were significant (*p* = 0.00888 and *p* = 0.00322, respectively; Welch’s *t*-test with Bonferroni correction). These results indicate that NMDA-R and PKA activation are necessary for LTP-STDP by pre-before-post timing stimulation, but that they are not sufficient conditions.

### 3.3. Distinct Induction Mechanism for LTP-STDP by Post-Before-Pre Timing Stimulation: It Requires PKA but Not NMDA-R

Subsequently, we examined the involvement of NMDA-R and PKA in the induction of LTP-STDP by post-before-pre timing stimulation. To our surprise, D-AP5 (50 μM) failed to block the induction of this type of LTP-STDP, as exemplified in Figure 3A. Nevertheless, PKA was found to be still necessary for LTP-STDP by post-before-pre timing stimulation, as shown in Figure 3B. These results are summarized in Figure 3C. The application of D-AP5 could not suppress the induction of LTP-STDP by post-before-pre timing stimulation (control; 125.7 ± 5.35%, n = 21, D-AP5; 123.3 ± 7.61%, n = 6). However, PKI 6-22 blocked its induction (100.1 ± 3.80%, n = 8). One-way ANOVA analysis indicated significant differences among groups (F(2,32) = 4.28, *p* = 0.0225). The following post hoc test indicated that the difference between control and D-APV was not significant, while that between control and PKI 6-22 was significant (*p* = 1.00 and *p* = 0.0012, respectively, Welch t-test with Bonferroni correction). These results indicate that LTP-STDP by the post-before-pre timing stimulation was produced by PKA activation, without the activation of NMDA-R. Thus, LTP-STDP by the post-before-pre timing stimulation exhibited distinct induction mechanism from that by the pre-before-post timing stimulation.

## 4. Discussion

### 4.1. All-LTP STDP at L2/3–L2/3 Horizontal Connection Between Neighboring Columns During Development

L2/3 horizontal connections exhibited all-LTP STDP throughout our observation (P9–P13). Previous studies showed that synapses exhibit all-LTP STDP during circuit formation and synaptogenesis, some examples of which include thalamocortical synapses at the first postnatal week [14] and L4–L2/3 connections at the second postnatal week [2,4]. Developmental studies demonstrated that the branch number and axon length of L2/3 pyramidal cells within the same layer from P7 to P22 increase throughout the observation period of P7–P22 but most rapidly from P7 to P15 [15]. Our observation of all-LTP STDP from P9 to P13 during the time of axon extension supports the idea that all-LTP STDP is the basis of activity-dependent circuit formation. It is predictable that this type of plasticity would be a common feature for immature synapses during synaptic formation at least in the cerebral cortex, but could also probably be in other synapses in the brain. It is of interest to test this hypothesis in other synapses beyond the cerebral cortex.

### 4.2. Involvement of PKA and NMDA-R

We demonstrated that all-LTP STDP at L2/3 horizontal connection between neighboring columns was mediated by PKA. This is again consistent with all-LTP STDP at thalamocortical and at L4-L2/3 synapses during synapse formation as we demonstrated [4,14]. At the L4-L2/3 synapses, all-LTP STDP switches to normal Hebbian STDP with LTP by the pre-before-post timing stimulation and LTD by the post-before-pre timing stimulation, and this LTP is mediated by CaMKII [4]. Thus, even the same LTP is produced, but with distinct signaling mechanism. A similar developmental switch of signaling cascade in LTP has also been reported in the hippocampus where hippocampal LTP is mediated by PKA-dependent GluR4 delivery to the membrane in early postnatal weeks [27,28]. These observations are consistent with the finding that the amount of CaMKII and GluR1 is developmentally regulated and is hardly abundant in the forebrain until the third postnatal week [28,29]. S1 map plasticity is considered to have two separable components [1,2]; first, the selective weakening of neural responses to deprived whiskers and second, an enhancement of responses to spared whiskers in the deprived columns [5,6,7], which has been suggested to result from enhanced excitatory synaptic connections among cross-columnar projections from spared to deprived columns. A strong cellular candidate for this process would be L2/3 horizontal cross-columnar synapses. Thus, it is of great interest to see if these L2/3 horizontal cross-columnar synapses exhibit LTP by either of both pre-before-post and/or post-before-pre timing stimulations after the initiation of the critical period after P15, and suppose LTP is produced, it is also of interest to find if it is mediated by PKA or CaMKII.

### 4.3. Involvement of NMDA-R

NMDA-R was required for LTP by the pre-before-post timing stimulation, but was not required for that by the post-before-pre timing stimulation. Previous studies demonstrated that PKA signaling pathway can be triggered by calcium entry through NMDA-R, mediated by calcium-sensitive adenylyl cyclase [30,31], and in fact, NMDA-R-mediated PKA signaling underlies GluR4 delivery to synaptic sites in immature hippocampal neurons [27]. In contrast, LTP by post-before-pre timing stimulation did not require NMDA-R activation. Then, what is the upstream mechanism of PKA activation? Considering that L2/3-L2/3 synapses are glutamatergic, an authentic activation of upstream Gs pathway such as through adrenergic β or glucagon receptors is unlikely; thus, the entry of Ca^2+^ from glutamate receptors other than NMDA-R is more likely. Possible candidates would include mGluR and GluR2-lack AMPA receptors, but voltage-gated Ca^2+^ channels should also not be excluded. We hope to identify the correct Ca^2+^ source and report it in the near future.

## Figures and Tables

**Figure 1 cells-14-01459-f001:**
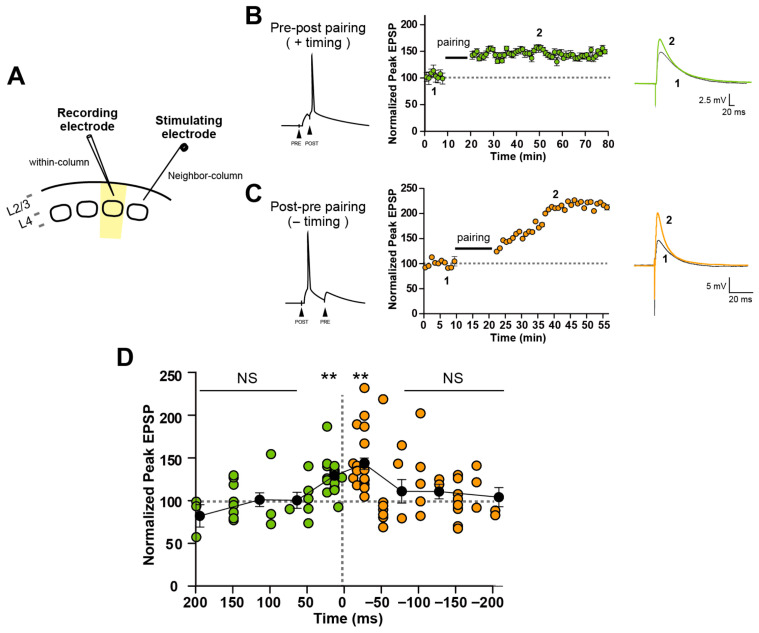
All-LTP STDP was induced by both pre-before-post as well as post-before-pre timing stimulations at L2/3-L2/3 horizontal connections between neighboring columns during developmental immature synapse formation from P9-13. (**A**) Schematic illustration. Recording was made from L2/3 pyramidal cells in response to the stimulation of the neighboring column. (**B**) Left: postsynaptic action potential was induced after the onset of EPSP caused by the stimulation of a presynaptic input. Middle: an example plot of normalized EPSP amplitudes against time. Right: averaged (10 sweeps) EPSPs obtained from the numbered timing in the graph. (**C**) Left: postsynaptic action potential was induced before the onset of EPSP caused by the stimulation of presynaptic input. Middle: an example plot of normalized EPSP amplitudes against time. Right: averaged (10 sweeps) EPSPs obtained from the numbered timing in the graph. (**D**) The graph shows the relationship between applied timing delay from 200 ms (pre-before-post) to −200 ms (post-before-pre) and the resultant changes in the normalized EPSP amplitude. Mean ± SEM were calculated from the data for the timing of +200 ms, +150~+100 ms, +75~+50 ms, +25~+5 ms, −10~−25 ms, −50~−75 ms, −100~−150 ms, and −175~−200 ms, then indicated in the graph. NS; not significant, ** *p* < 0.01, Student *t*-test, n = 89.

**Figure 2 cells-14-01459-f002:**
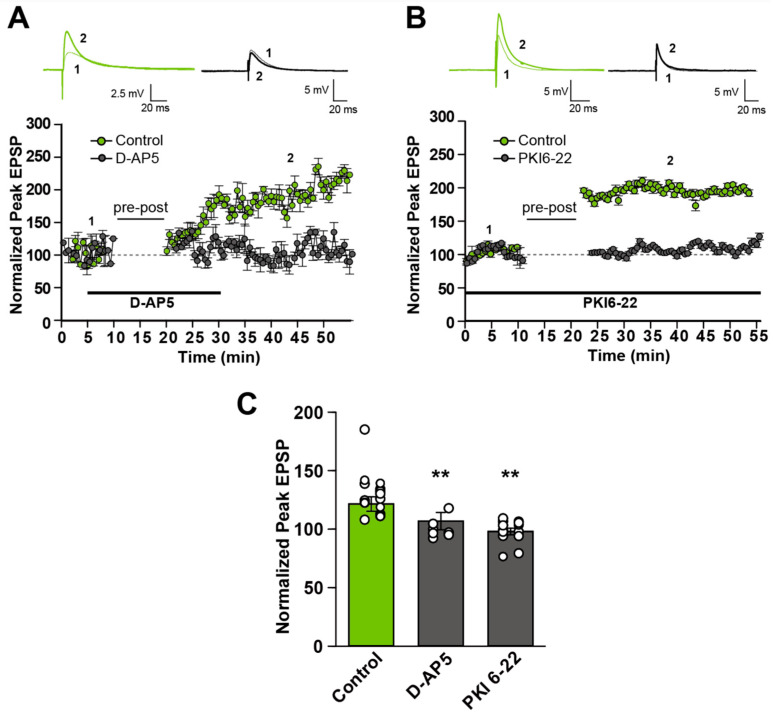
Effects of NMDA-R antagonist and PKA inhibitor on STDP induction by pre-before-post (+15 ms delay) timing stimulation. (**A**,**B**) An example plot (bottom) of the effect of NMDA-R antagonist, D-AP5 (**A**) and PKA inhibitor, PKI6-22 (**B**). Example recordings are shown at the top, obtained at the designated timing in the graph. (**C**) Quantification of the effect of D-AP5 and PKI6-22 from the pooled data. ** *p* < 0.01 (One-way ANOVA followed by Welch’s *t*-test with Bonferroni correction).

**Figure 3 cells-14-01459-f003:**
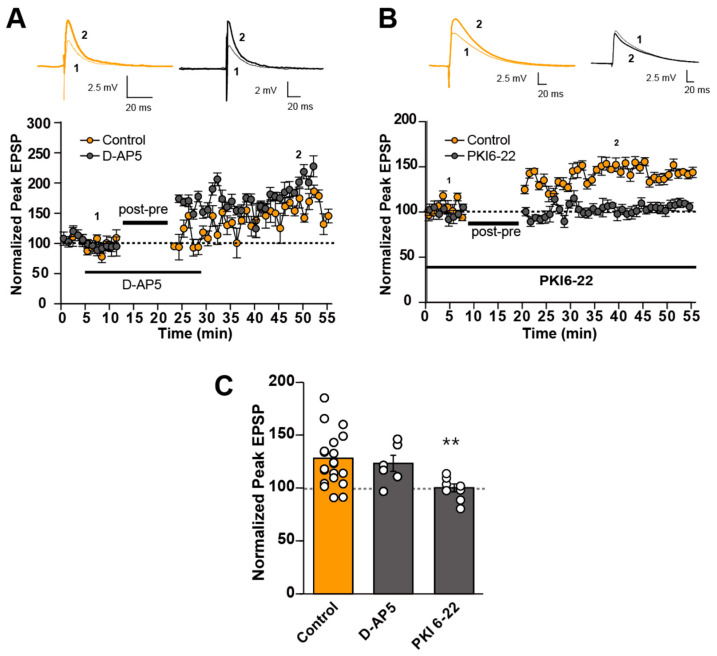
Effects of NMDA-R antagonist and PKA inhibitor on STDP induction in post-before-pre (−25 delay) timing stimulation. (**A**,**B**) An example plot (bottom) of the effect of NMDA-R antagonist, D-AP5 (**A**) and PKA inhibitor, PKI6-22 (**B**). (**C**) Quantification of the effect of D-AP5 and PKI6-22 from the pooled data. ** *p* < 0.01 (One-way ANOVA followed by Welch’s *t*-test with Bonferroni correction).

## Data Availability

The original contributions presented in this study are included in the article. Further inquiries can be directed to the corresponding authors.

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
