# Peer review of "Spike Timing-Dependent Plasticity at Layer 2/3 Horizontal Connections Between Neighboring Columns During Synapse Formation Before the Critical Period in the Developing Barrel Cortex"

_cells, 2025, doi:10.3390/cells14181459_

Round 1

Reviewer 1 Report

Comments and Suggestions for Authors

In this manuscript, the authors investigate the role of spike timing-dependent plasticity (STDP) in the horizontal connections between layer 2/3 (L2/3) pyramidal cells in adjacent columns of the rodent barrel cortex during development. Specifically, they examine all-LTP STDP, a form of plasticity in which both pre-before-post and post-before-pre-spike timing lead to long-term potentiation (LTP), prior to the critical period of sensory map plasticity.

The study focuses on the period from postnatal day 9 to 13 (P9–P13), during the formation of synapses in L2/3 neurons. Electrophysiological recordings show that all-LTP STDP occurs during this period, with both pre-before-post and post-before-pre-timing leading to significant LTP, especially at shorter delays. The results suggest that L2/3 horizontal connections between neighboring columns exhibit all-LTP STDP at least from P9, well before the critical period at P15 when more mature forms of STDP, including both LTP and LTD, become prominent.

Furthermore, the authors explore the molecular mechanisms involved in this form of LTP-STDP. They find that NMDA receptors (NMDA-R) are required for LTP-STDP induced by pre-before-post stimulation, while protein kinase A (PKA) signaling is necessary for both pre-before-post and post-before-pre LTP induction. However, for LTP-STDP induced by post-before-pre-stimulation, NMDA-Rs are not involved, suggesting a distinct mechanism for this form of STDP, which still relies on PKA signaling.

The study concludes that L2/3 horizontal connections begin to undergo all-LTP STDP at least by P9, during synapse formation, and before the critical period of map plasticity in the barrel cortex, highlighting the importance of PKA signaling in this early stage of cortical circuit development.

Overall, this is a clearly presented study that provides high-quality electrophysiology data. I only have a few comments, which are detailed below:

  1. The authors may have included this information, but it was not clear what delay was used between pre-post and post-pre stimulation when collecting the data shown in Figs. 2 and 3.

Minor:

  1. Lines 169, 171, and 186 the micro in micromolar is missing.
  2. Line 223, misplaced period before, “and.”

Author Response

In this manuscript, the authors investigate the role of spike timing-dependent plasticity (STDP) in the horizontal connections between layer 2/3 (L2/3) pyramidal cells in adjacent columns of the rodent barrel cortex during development. Specifically, they examine all-LTP STDP, a form of plasticity in which both pre-before-post and post-before-pre-spike timing lead to long-term potentiation (LTP), prior to the critical period of sensory map plasticity.

The study focuses on the period from postnatal day 9 to 13 (P9–P13), during the formation of synapses in L2/3 neurons. Electrophysiological recordings show that all-LTP STDP occurs during this period, with both pre-before-post and post-before-pre-timing leading to significant LTP, especially at shorter delays. The results suggest that L2/3 horizontal connections between neighboring columns exhibit all-LTP STDP at least from P9, well before the critical period at P15 when more mature forms of STDP, including both LTP and LTD, become prominent.

Furthermore, the authors explore the molecular mechanisms involved in this form of LTP-STDP. They find that NMDA receptors (NMDA-R) are required for LTP-STDP induced by pre-before-post stimulation, while protein kinase A (PKA) signaling is necessary for both pre-before-post and post-before-pre LTP induction. However, for LTP-STDP induced by post-before-pre-stimulation, NMDA-Rs are not involved, suggesting a distinct mechanism for this form of STDP, which still relies on PKA signaling.

The study concludes that L2/3 horizontal connections begin to undergo all-LTP STDP at least by P9, during synapse formation, and before the critical period of map plasticity in the barrel cortex, highlighting the importance of PKA signaling in this early stage of cortical circuit development.

Overall, this is a clearly presented study that provides high-quality electrophysiology data. I only have a few comments, which are detailed below:

  1. The authors may have included this information, but it was not clear what delay was used between pre-post and post-pre stimulation when collecting the data shown in Figs. 2 and 3.

Response to comment 1

We appreciate this reviewer for his/her affirmative comments.

1. We are grateful for the reviewer’s point. We used +15 ms delay for pre-before-post and -25ms delay for post-before-pre timing stimulation, and thus we have explicitly described the delay timing in the figure legend.

Reviewer 2 Report

Comments and Suggestions for Authors

In this paper the authors set out to examine the role of STDP plasticity and the mechanistic role of NMDA-Rs and PKA in the phenomenon where after whisker trimming, responses in the speared whisker are potentiated. While I appreciate the attempt to link plasticity and the mechanistic study to an in vivo phenomenon, this manuscript has no experiments where whiskers are trimmed, so the link between the findings and the phenomenon the authors claim to be interested in is theoretical at best. Aside from this conceptual issue, the paper has some methodological problems, including the lack of quality control and inappropriate statistical design. All these issues are fixable, and the paper is interesting enough to warrant publication assuming that the findings hold after removing poor quality recordings from the dataset and using appropriate statistical comparison.

Major issues

  1. Cimbing baseline. Figure 1B contains an example time series recording where the baseline is clearly increasing at a rate that would suggest similar amplitude after 15 minutes as the one displayed without LTP induction. Similar climbing baseline is evident in figures 2 and 3, suggesting that Figure 1B is not an outlier but the result of weak quality control. All recordings with unstable baseline should be removed from analysis. In its current state, it cannot be clearly stated that all displayed (and quantified) recordings show LTP.
  2. To be able to claim that the recordings indeed reflect effects on horizontal connections, the authors need to describe stimulating electrode placement and how they determined that no axons of passage were stimulated by their method. If non-specific stimulation cannot be ruled out, this caveat must be discussed.
  3. Statistical issues: the authors compare control recordings to both NMDA-R and PKA block, but the comparisons are set up as separate unpaired t-tests. This data requires a one-way ANOVA followed by post-hoc tests corrected for multiple comparisons. Without this, the hypotheses are incorrectly tested and the results cannot be interpreted.
  4. Conceptual over-reach. Claiming that the paper studies the mechanisms of “spared whisker response potentiation” is a bit of a leap, considering that this study is done during early development (during critical period) while that phenomenon is often studied in adults. Additionally, there is no attempt in the manuscript to trim whiskers or relate to the in vivo phenomenon in any way. I would recommend adjusting the language to make it clear that the study focuses purely on STDP mechanisms.

Minor issues

  1. Why are the peak EPSPs in the control groups different between the NMDA experiment (~150, after normalization) and the PKA experiment (~200, after normalization) following induction?
  2. The authors should justify why was D-AP5 applied starting 5 minutes before induction and maintained until approximately 10 minutes after induction while PKI6-22 was applied continuously, including during the baseline recording and after induction
  3. Results section contains significant literature review. While I have no problem with a few references to place the experiments in context and increase readability, what we have here is a bit excessive. Consider moving some of it into the introduction or discussion.
  4. In Lines 175–176, the statement “These results indicate that LTP-STDP by pre-before-post timing stimulation was caused by NMDA-R and PKA activation” may be too strong. Based on the data, it is more accurate to conclude that NMDA-R and PKA activation are necessary for LTP-STDP, but not that they are sufficient conditions.
  5. In the methods the authors indicate that they conducted experiments at P9 and at P13 but we see no indication later in the paper about these ages: were there no differences so the data was pooled? How was this tested?
  6. Figure 1 sample sizes are not clearly indicated
  7. There seem to be numerous typos (or conversion errors) with concentration units.
  8. In Figure 1D, the unit on the x-axis should be “ms” rather than “min.”
  9. There appears to be an unusually large space in Line 247 that should be corrected.

Author Response

Comment of Reviewer 2

In this paper the authors set out to examine the role of STDP plasticity and the mechanistic role of NMDA-Rs and PKA in the phenomenon where after whisker trimming, responses in the speared whisker are potentiated. While I appreciate the attempt to link plasticity and the mechanistic study to an in vivo phenomenon, this manuscript has no experiments where whiskers are trimmed, so the link between the findings and the phenomenon the authors claim to be interested in is theoretical at best. Aside from this conceptual issue, the paper has some methodological problems, including the lack of quality control and inappropriate statistical design. All these issues are fixable, and the paper is interesting enough to warrant publication assuming that the findings hold after removing poor quality recordings from the dataset and using appropriate statistical comparison.

Major issues

  1. Cimbing baseline. Figure 1B contains an example time series recording where the baseline is clearly increasing at a rate that would suggest similar amplitude after 15 minutes as the one displayed without LTP induction. Similar climbing baseline is evident in figures 2 and 3, suggesting that Figure 1B is not an outlier but the result of weak quality control. All recordings with unstable baseline should be removed from analysis. In its current state, it cannot be clearly stated that all displayed (and quantified) recordings show LTP.
  2. To be able to claim that the recordings indeed reflect effects on horizontal connections, the authors need to describe stimulating electrode placement and how they determined that no axons of passage were stimulated by their method. If non-specific stimulation cannot be ruled out, this caveat must be discussed.
  3. Statistical issues: the authors compare control recordings to both NMDA-R and PKA block, but the comparisons are set up as separate unpaired t-tests. This data requires a one-way ANOVA followed by post-hoc tests corrected for multiple comparisons. Without this, the hypotheses are incorrectly tested and the results cannot be interpreted.
  4. Conceptual over-reach. Claiming that the paper studies the mechanisms of “spared whisker response potentiation” is a bit of a leap, considering that this study is done during early development (during critical period) while that phenomenon is often studied in adults. Additionally, there is no attempt in the manuscript to trim whiskers or relate to the in vivo phenomenon in any way. I would recommend adjusting the language to make it clear that the study focuses purely on STDP mechanisms.

Minor issues

  1. Why are the peak EPSPs in the control groups different between the NMDA experiment (~150, after normalization) and the PKA experiment (~200, after normalization) following induction?
  2. The authors should justify why was D-AP5 applied starting 5 minutes before induction and maintained until approximately 10 minutes after induction while PKI6-22 was applied continuously, including during the baseline recording and after induction
  3. Results section contains significant literature review. While I have no problem with a few references to place the experiments in context and increase readability, what we have here is a bit excessive. Consider moving some of it into the introduction or discussion.
  4. In Lines 175–176, the statement “These results indicate that LTP-STDP by pre-before-post timing stimulation was caused by NMDA-R and PKA activation” may be too strong. Based on the data, it is more accurate to conclude that NMDA-R and PKA activation are necessary for LTP-STDP, but not that they are sufficient conditions.
  5. In the methods the authors indicate that they conducted experiments at P9 and at P13 but we see no indication later in the paper about these ages: were there no differences so the data was pooled? How was this tested?
  6. Figure 1 sample sizes are not clearly indicated
  7. There seem to be numerous typos (or conversion errors) with concentration units.
  8. In Figure 1D, the unit on the x-axis should be “ms” rather than “min.”
  9. There appears to be an unusually large space in Line 247 that should be corrected.

Response to the Reviewer 2

We appreciate the valuable comments from this knowledgeable reviewer. We realized that our initial wording may have caused a misunderstanding of the aim of this study. Our intention is not to study spared whisker response potentiation, but rather to examine STDP during synaptic formation in the developing barrel cortex” as the title indicates (lines 2-4). Specifically, “all-LTP STDP when axons are extending” (lines 18-19, Abstract), which is “during synaptic formation at L2/3-L2/3 connection” (lines 24-25, Abstract), or “during network formation before critical period” (lines 49-50, Introduction). Furthermore, we explicitly described in the introduction that “In the present study, we tried to determine whether STDP could by induced …before the suggested age of the initiation of the critical period, that is, before P15” (lines 57-59). To explain the background a little more, exact age of the transition from neonatal network formation to the initiation of the critical period was proposed to be around P13-15 in the barrel cortex. We are interested in finding what type of STDP operates during the period of network formation before P13, the initiation of the critical period. That is the reason why we do not think that whisker trimming is necessary.

The reviewer claimed that the paper has some methodological problems, which are detailed in the “Major and Minor issues”, thus we respond to them in those sections below.

Major issues

  1. Climbing baseline.

We checked all the data and unstable recordings with baseline drift were excluded. Accordingly, we have replaced the example recordings and plots with more stable baselines.

  1. Stimulating electrode placement

We appreciate the reviewer’s comment, regarding the possibility of stimulating the passing fibers. We placed the stimulating electrode on the neighboring column at about the same cortical depth from the pia surface with the recorded cells (Fig. 1A). Since L2/3 pyramidal cells send their axons horizontally across columns, we were likely to stimulate L2/3 horizontal fibers, which include not only from cells in the next column, but also from further columns. Thus, we have added this caveat in the Method section (lines 113-117).

  1. Statistical issues

Following to the reviewer’s comment, Statistical tests have been performed using one-way ANOVA followed by post-hoc tests corrected for multiple comparison for the data Fig. 2 and 3. Related descriptions have been changed in Results (lines 180-184, 202-206)   and Figure legends (lines 191-192, 214-215).

  1. Conceptual over-reach.

As we described above, this study aimed to examine whether all-LTP STDP operates during synapse formation before the suggested age of the initiation of the critical period, and definitely not the mechanism of “spared whisker response potentiation.” Thus, we have not changed the manuscript in this regard.

Minor issues

  1. The reviewer referred to the previous Fig. 2 testing the D-APV and PKI 6-22. These are example of single cells and the amount of LTP differs from one cell to another, as shown in Fig. 1D. Anyway, these examples have been replaced with more stable baselines, though.

  1. This is because PKI 6-22 was applied intracellularly by pipette solution, while D-AP5 was applied externally from bath solution.

  1. According to the reviewer’s suggestion, 6 references have been moved to introduction (lines 62-69).

  1. The statement has been changed according to the reviewer’s suggestion and have been replaced in lines 184-185.

  1. We collected data from P9 to P13. The number of cells for each age are as follows.

P9, n=15; P10, n=5; P11, n=18; P12, n=21; P13, n=26.

For timing stimulation, the delay parameter between pre and post stimulation we used are eleven, such as

-200 ms, -150 ms, -100 ms, -75 ms, -50 ms, -25 ms, +15 ms, +50 ms, +100 ms, +150 ms, +200 ms

For eleven different parameters, we think it is difficult to discuss statistically the age-dependent differences for five different days using a total of 85 cells. Thus, we pooled all the data, as an age group of “before the critical period”.

  1. Figure 1, B and C are representative single data for pre-post (B) and post-pre (C) stimulation. In D, each point represent single cells, so the total number of cells is 85 cells. which has been added in Figure legend (line 164)

  1. These are conversion errors occurred when our manuscript was converted to the form of the Journal. All the errors have been corrected.

  1. In Figure 1D, the unit on the x-axis has been corrected.

  1. Apparently the large space that the reviewer pointed has been already corrected by Journal staff.

Round 2

Reviewer 2 Report

Comments and Suggestions for Authors

The authors addressed some of the issues we brought up, however, some problems remain.

  1. In response to comments, the authors claim that unstable and low-quality recordings were removed from the analysis. It is evident from the change in the example traces shown in every figure that some recordings were removed. Yet, the sample size in all statistics and the population data in the bar graphs remain completely unchanged from the first submission (even p-values are the same). So did the authors just change the example traces but did not remove poor quality data from the quantification? This makes no sense. Data not good enough to show is certainly not good enough to be in the statistics and population data.

  1. I understand the authors’ explanation of the study’s goals. The problem is that they don’t need to explain this to me, they need to make this clear in the manuscript. As it stands, the language is not clear.

  1. The authors explain in their response that the PKA inhibitor PKI 6-22 was applied intracellularly via the patch pipette. Why is this not mentioned anywhere in the manuscript?

Author Response

Reviewer's point 1

 In response to comments, the authors claim that unstable and low-quality recordings were removed from the analysis. It is evident from the change in the example traces shown in every figure that some recordings were removed. Yet, the sample size in all statistics and the population data in the bar graphs remain completely unchanged from the first submission (even p-values are the same). So did the authors just change the example traces but did not remove poor quality data from the quantification? This makes no sense. Data not good enough to show is certainly not good enough to be in the statistics and population data.

Response to 1

We seemed to mistakenly use unrevised text partially in the previous revisison. We have actually discarded some data with poor quality and further added some more data. Thus the total number of cells has increased from 85 to 89, which has been added in Figure 1 legend (line171). Accordingly Figure 2C and Figure 3C have been changed. In addition, we described the criteria for data in the Method (line 124-127).

Reviewer's point 2

I understand the authors’ explanation of the study’s goals. The problem is that they don’t need to explain this to me, they need to make this clear in the manuscript. As it stands, the language is not clear.

Response to 2

We originally described “before the critical period” (2 places)  and “during synapse formation” (7 places) in the text to make clear the goal of our study. According to the reviewer’s suggestion, we haved further added “before the critical period” in the title (line 4) and abstract (line 20).

Reviewer's point 3

The authors explain in their response that the PKA inhibitor PKI 6-22 was applied intracellularly via the patch pipette. Why is this not mentioned anywhere in the manuscript?

Response to 3

We have added how these drugs were applied in the text (lines 182, 184) for clarity.

Round 3

Reviewer 2 Report

Comments and Suggestions for Authors

How drugs were applied should be in the methods section, not as an addition in the main text, otherwise the revisions are acceptable.

Author Response

Comment

How drugs were applied should be in the methods section, not as an addition in the main text, otherwise the revisions are acceptable.

Response

We have added how drugs were applied in the methods section (lines 133-137).